# Bisdemethoxycurcumin Alleviates Dextran Sodium Sulfate-Induced Colitis via Inhibiting NLRP3 Inflammasome Activation and Modulating the Gut Microbiota in Mice

**DOI:** 10.3390/antiox11101994

**Published:** 2022-10-07

**Authors:** Jingfei Zhang, Qiming Li, Xin Zhang, Yanan Chen, Yufang Lu, Xinyu Wang, Lili Zhang, Tian Wang

**Affiliations:** College of Animal Science and Technology, Nanjing Agricultural University, Nanjing 210095, China

**Keywords:** bisdemethoxycurcumin, colitis, inflammatory response, intestinal barrier function, apoptosis, NLRP3, gut microbiota

## Abstract

Our previous study showed that bisdemethoxycurcumin (BUR) exerts anti-inflammatory properties in lipopolysaccharide-induced intestinal injury, and studies have revealed that NOD-like receptor superfamily, pyrin domain containing 3 (NLRP3) inflammasome activation plays a vital role in the pathogenesis of colitis. However, it is not clear whether BUR could attenuate colitis-mediated intestinal inflammation via NLRP3 inflammasome inactivation and modulate the gut microbiota dysbiosis. The results demonstrated that BUR attenuated DSS-induced body weight decrease, histopathological changes, and epithelial apoptosis. BUR significantly improved the intestinal barrier defects and abrogated DSS-induced inflammatory response. Consistently, BUR reduced the expression of NLRP3 family members, confirming its inhibitory effects on NLRP3 inflammasome activation and pyroptosis. BUR regulated microbiota dysbiosis and altered the gut microbial community. BUR supplementation enriched the relative abundance of beneficial bacteria (such as *Lactobacillus* and *Bifidobacterium*), which showed significant negative correlations with the pro-inflammatory biomarkers. Collectively, these findings illustrated that BUR could ameliorate DSS-induced colitis by improving intestinal barrier function, reducing apoptosis, inhibiting NLRP3 inflammasome activation, and regulating the gut microbiota.

## 1. Introduction

With the prevalence of irregular eating patterns and unfriendly environmental factors, inflammatory bowel diseases (IBD) have increased annually, including ulcerative colitis and Crohn’s disease, and have become one of the major threats to health globally [1]. The pathogenesis of IBD remains unclear, but recently researchers have confirmed that excessive oxidative stress plays an important role in the development of IBD [2]. Reactive oxygen species-related cell apoptosis and inflammasome activation aggravate intestinal barrier dysfunction and promote the progression of gut microbiota dysbiosis [3]. Dietary antioxidant intervention not only alleviates oxidant stress in inflamed intestine, but also regulates the immune−mediated inflammatory disorders [4,5,6]. In dextran sulfate sodium (DSS)−induced colitis, a well−accepted mouse model of IBD, natural antioxidants are reported to suppress inflammatory disorders through inflammasome inactivation and inhibition of inflammation. Therefore, supplementation with natural antioxidants may be a novel alternative treatment and intervention strategy for IBD. 

Inflammasome is an important component of innate immune response, and the NOD−like receptor superfamily, pyrin domain containing 3 (NLRP3) inflammasome is one of the most characterized inflammasomes. The NLRP3 inflammasome consists of a sensor molecule NLRP3, the adaptor apoptosis−associated speck−like protein containing a CARD (ASC), and the effector protease caspase 1 [7]. The NLRP3 inflammasome is activated by the oligomerization of NLRP3 with the ASC and pro−caspase 1 in a canonical pathway, which leads to the cleavage of caspase 1 and subsequently, the maturation of interleukin−1beta (IL−1β) and interleukin−18 (IL−18) [8]. These NLRP3−mediated inflammatory cascades are responsible for intestinal injury in response to inflammation [9]. Bauer and colleagues reported that NLRP3 contributed to inflammatory bowel disease in DSS−triggered colitis and was detrimental for intestinal epithelial barrier−maintenance [10]. Moreover, NLRP3 triggers pyroptosis via the N−terminal fragment of gasdermin−D (GSDMD), which forms pores in the plasma membrane, induces cell member rupture, and enhances an inflammatory response [11]. Recent studies showed that NLRP3 inactivation by siRNA or a specific inhibitor efficiently protected against intestinal barrier dysfunction and apoptosis, and alleviated colitis−induced cells’ pyroptosis [10,12]. Therefore, the inhibition of the NLRP3 inflammasome and NLRP3−initiated pyroptosis is a promising therapeutic strategy for the prevention and treatment of the inflamed intestine in colitis. 

Curcumin is a bioactive polyphenolic compound of turmeric (*Curcuma longa*), and widely used for traditional medicine in Indian and other Asian countries. Besides its excellent antioxidant activity, curcumin is reported to alleviate a wide spectrum of acute and chronic inflammatory disorders. Studies found that curcumin could suppress NLRP3 inflammasome activation in primary microglia and protect mice against ischemic stroke [13]. It was also indicated that curcumin alleviated DSS−induced colitis via inhibiting NLRP3 inflammasome activation and IL−1β production [14]. In doxorubicin−induced mice, curcumin inhibited NLRP3−mediated pyroptosis and enhanced the innate immune system [15]. 

Despite its excellent anti−inflammatory properties, there is limitation for the use of curcumin due to its poor absorption and bioavailability in vivo. Bisdemethoxycurcumin (BUR) is a derivative of curcumin and has more efficient pharmacological properties than curcumin [16]. In recent years, BUR has attracted increasing attention owing to its better bioavailability in vivo when compared with curcumin [17,18]. BUR is a minor constituent of curcuminoids and shows antioxidant, anti−inflammatory, antimutagenic, and antitumor effects [19]. Many studies have demonstrated that BUR has positive effects on organ protection, such as the liver, intestines, and kidneys. Our previous study reported that BUR was more potent than curcumin in alleviating circulating lipid peroxidation and facilitating the antioxidant gene expression of the liver and small intestine in broilers [20]. The latest research from our laboratory certified the protection of BUR against LPS-induced intestinal injury via reducing the inflammatory response. In a subsequent nutritional intervention study, we observed a decreased mRNA expression of intestinal IL-1β in broilers [21]. Moreover, it inhibited the neuroinflammation via the inhibition of the AKT/NF-κB and p38/JNK pathway in LPS-induced RAW 264.7 macrophage cells [22], and enhanced reno-protective effects and attenuated the toxicity of cisplatin in HK-2 cells and ICR mice [23]. However, whether BUR reduces IL-1β production via inhibiting NLRP3 inflammasome activation, and thereby attenuating inflammation has not yet been revealed. 

In this study, we established a DSS-induced intestinal inflammatory model in C57BL/6 mice to explore the effects of BUR in suppressing intestinal barrier dysfunction, moderating intestinal epithelial cell apoptosis, inhibition inflammatory response, and altering gut microbiota dysbiosis. Furthermore, the inhibitory effects of BUR on NLRP3 inflammasome activation and pyroptosis were further investigated. 

## 2. Materials and Methods

All animal procedures were performed according to the guiding principles of the Animals Care and Ethics Committee of Nanjing Agricultural University, China (Certification No.: SYXK (Su) 2017–0007, 29 June 2021).

### 2.1. Animal Treatment

A total of 45 6−week−old male C57BL/6 mice (18–21 g) were purchased from Shanghai SLAC Laboratory Animal Ltd. Co. (Shanghai, China). The mice were caged in an environmentally controlled room (24 ± 1 °C, 12 h light–dark cycle and 50 ± 10% humidity) and had free access to water and food. Mice were randomly divided into four groups (six animals per group): the control group; DSS group; BUR (200 mg/kg) group; and BUR (400 mg/kg) group. The experimental timelines are shown in Figure 1A. Mice received corn oil, either 200 or 400 mg/kg body weight by gavage once per day for 14 days. On day 7–14, mice in the DSS and BUR groups were receiving DSS (2.5%) in drinking water to establish the colitis model. The dosage of DSS was chosen based on the previous reports [24,25]. The BUR used in the present study was provided by Kehu Bio−technology Research Center (purity ≥ 98%, Guangzhou, China). The doses of 200 and 400 mg/kg BUR used in the present study were according to our previous studies [21,24,26]. 

On day 15, all mice were sacrificed, and the serum and colons were collected. The colons were photographed, and the lengths of colons were immediately measured. Then, a small part of the colon was fixed in 4% paraformaldehyde for histological analysis; the remainder were snap frozen in liquid nitrogen and stored at −80 °C for subsequent analysis.

### 2.2. Disease Activity Index (DAI)

During the experimental period, the body weight, stool consistency, and fecal occult blood were recorded daily. The DAI score was calculated by the weight loss, stool consistency, and fecal occult blood as described previously (Appendix A) [27].

### 2.3. Determination of Inflammatory Cytokine

Interleukin−6 (IL−6), IL−1β, tumor necrosis factor−alpha (TNF−α), IL−18 in serum were measured using enzyme−linked immunosorbent assay kits (MultiSciences (Lianke) Biotech Co., Ltd., Hangzhou, China) according to the manufacturer’s instructions.

### 2.4. Histological Analysis

The fixed colon specimens were embedded in paraffin and cut into 5 μm slices. Then, the colon slices were stained with hematoxylin–eosin (H&E, Servicebio, China) and Alcian−blue/periodic acid Schiff reagent (AB−PAS, Servicebio, China) and observed using a light microscope.

### 2.5. Determination of Myeloperoxidase (MPO) Activity

The colonic MPO activity was measured using a commercial kit (Nanjing Jiancheng Bioengineering Institute, Nanjing, China) according to the manufacturer’s instructions.

### 2.6. Terminal Deoxynucleotidyl Transferase−Mediated dUTP Nick End Labeling (TUNEL) Assay

The apoptosis of the colon was detected by TUNEL assay using a TUNEL Bright Green apoptosis detection kit (Vazyme, Nanjing, Jiangsu, China). Briefly, the paraffin sections of the colon were dewaxed with xylene and rehydrated with ethanol. After treating with a proteinase K (20 μg/mL) for 20 min at room temperature, the tissue was incubated with 50 μL TdT buffer for 60 min at 37 °C in a dark and wet environment. Nuclei were then labeled with 4,6−di−amidino−2−phenylindole (Beyotime, Shanghai, China). Images were acquired using a confocal laser scanning microscope (Carl Zeiss) and analyzed by the Image−Pro Plus 6.0 software. The results were expressed as the ratio of the number of TUNEL−positive cells (green cell) to the total cell (blue cell).

### 2.7. Real Time−PCR (RT−PCR) Analysis

Total RNA from colon samples were isolated using TRIzol Reagent (Takara, Dalian, China). The extracted RNA was calculated using a NanoDrop ND−1000 UV spectrophotometer (NanoDrop Technologies, Thermo Scientific, Waltham, MA, USA) at 260 and 280 nm and reverse−transcribed to cDNA using a PrimeScript RT Reagent kit (Takara Biotechnology Co., Dalian, China). The cDNA samples were amplified using the SYBR^®^ Premix Ex Taq™ Kit (Takara Biotechnology Co. Ltd., Dalian, China) and a QuantStudio^®^5 real−time PCR Design & Analysis system (Applied Biosystems, USA) according to the manufacturer’s instructions. Briefly, the primer sequences are listed in Appendix A. The relative expression levels were calculated with the 2 ^−∆∆Ct^ method. β−action was chosen as the internal control and used to normalize the results of target genes as described previously [24].

### 2.8. Western Blot Analysis

The colon tissues were homogenized with the ice−cold RIPA buffer containing protease inhibitors (Beyotime Biotechnology, Shanghai, China). The colon proteins were collected after centrifugation at 14,000× *g* for 5 min and determined using BCA protein assay kit (Beyotime Biotechnology, Shanghai, China). Equal amounts of protein were separated by 4–12% SDS−PAGE gel and transferred to polyvinylidene difluoride membranes (Millipore, Bedford, MA, USA). After blocking at room temperature with a commercial blocking buffer (Beyotime Biotechnology, Shanghai, China), membranes were incubated overnight at 4 °C with the following primary antibodies: Caspase 3, B cell lymphoma2 (Bcl2), Bcl2 associated X (Bax), cleaved Caspase 3 P17, cleaved Caspase 1 P20, and cleaved IL−1β, which were purchased from Affinity Biosciences (Cincinnati, OH, USA); Occludin, Claudin 1, ZO−1, ACS, NLPR3, Caspase 1, IL−1β, GSDMD, and β−actin, which were purchased from Proteintech (Wuhan, Hubei, China). Then, membranes were incubated with HRP−conjugated secondary antibody for 2–2.5 h at room temperature. Finally, the protein bands were visualized using an enhanced chemiluminescence (ECL) kit (Thermo Scientific, Wilmington, DE, USA) and detected by the ChemiDoc™ imaging system (BIO−RAD, Hercules, CA, USA). The results were quantified using Image J software and normalized to β−actin.

### 2.9. Gut Microbiota Analysis

Total genomic DNA from twenty−four cecal digesta samples were extracted using the QIAamp Fast DNA Stool Mini Kit (Qiagen, MD, USA) according to the protocol. A PCR reaction was performed to amplify the V3–V4 region of the bacteria 16S rDNA using the specific primers with the barcode as previously described [21]. The PCR products were purified using the QIAquick Gel Extraction Kit (Qiagen, Germantown, MD, USA) and quantified using the QuantiFluor−ST Fluorometer (Promega, Madison, WI, USA). Then, sequencing was conducted with an Illumina Miseq PE300 platform (BIOZERON, Shanghai, China). 

According to the Mothur process (Ver 1.32.0, Ann arbor, MI, USA), data filtration was conducted to obtained qualified sequences. Sequences with a similarity of 97% were assigned to cluster the same operational taxonomic units (OTUs). Alpha diversity was analyzed to estimate the abundance and diversity of the bacterial community, including the Chao1, Shannon, and Simpson indices. Beta diversity analysis was performed by principal co−ordinate analysis (PCoA). Subsequent analysis of linear discriminant analysis effect size (LEfSe) was conducted to identify the significant and unique OTUs among groups.

### 2.10. Quantification of Short−Chain Fatty Acids (SCFAs)

The colonic digesta were homogenized with 1 mL distilled water and 0.2 μL metaphosphoric acid (25%, v/v). The mixture was stored at −20 ℃ for 30 min and then centrifuged for 10 min at 12,000 rpm. The supernatant was mixed vigorously with an equal volume of ice−cold ethyl ether and placed for 5 min at 4 ℃. The organic layers of samples were filtered with a 0.22 μm organic membrane and analyzed by gas chromatography as described previously [28,29]. 

### 2.11. Statistical Analysis

The data were expressed as mean ± standard error mean (SEM). Comparisons among groups were conducted by one−way analysis of variance (ANOVA) followed by Tukey’s post−hoc test or the Kruskal–Wallis test with Dunn’s post−hoc test using the SPSS 22.0 software. Spearman’s rank correlation test was performed to evaluate the correlations between the relative abundance of top 10 bacteria at genus level and inflammatory−related mediators. A value of *p*< 0.05 was considered statistically significant.

## 3. Results

### 3.1. BUR Alleviated DSS−Induced Colitis

To determine the effects of BUR on DSS−induced colitis, mice were treated by gavage with the control corn oil, 200 or 400 mg/kg BUR, respectively. During the 14−day experiment period, there was no mortality in any of the treatments. Weight loss after DSS treatment was significantly prevented by BUR (400 mg/kg) treatment in mice, but not by BUR (200 mg/kg) treatment (Figure 1B). BUR treatment significantly rescued the DSS−induced decrease in colon length (Figure 1C). Consistent with the data on body weight and colon length, BUR treatment significantly decreased the DAI scores (Figure 1D), suggesting the alleviated clinical colitis symptoms by BUR treatment. Additionally, serum IL−1β, TNF−α, IL−6, and IL−18 levels were analyzed. BUR pretreatment significantly reduced the levels of these inflammatory cytokines (Figure 1E–H). 

### 3.2. BUR Improved Intestinal Barrier Function

Histological examination showed that the damaged crypts, increased infiltration of inflammatory cells, and goblet cells’ loss were observed in DSS group, were improved by BUR treatment (Figure 2A,B). BUR pretreatment significantly decreased the MPO activity of colon, confirming the anti−inflammatory effects of BUR in DSS−induced colitis. 

To further investigate whether BUR protected the intestinal barrier function, we measured the mRNA expression of tight junction proteins. The results showed that DSS suppressed the mRNA expression of occludin, claudin−1, and ZO−1, which were prevented by BUR treatment (Figure 2E–G). Furthermore, Western blotting demonstrated similar changes in protein levels, indicating that BUR could efficiently improve DSS−induced intestinal barrier integrity in mice (Figure 2H). These data suggested that BUR improved the intestinal barrier function via regulating the tight junction proteins.

### 3.3. BUR Attenuated Apoptosis in Colon

Apoptosis is one of the mechanisms contributing to intestinal epithelial barrier dysfunctions and inflammatory response in DSS−induced colitis. We firstly determined whether BUR could attenuate apoptosis (Figure 3). According to the TUNEL assay, BUR pretreatment led to a significant decrease in the number of TUNEL−positive cells as compared to the DSS group (Figure 3A,B). We then confirmed whether the suppressive effect of BUR on apoptosis was mediated in a mitochondrial−dependent apoptotic pathway. The RT−PCR analyses showed that BUR pretreatment significantly suppressed the mRNA expression of Bax and caspase 3 while it promoted the mRNA expression of Bcl2 in DSS−treated mice. These observations were consistent with the Western blots’ results demonstrating that the protein expression of cleaved caspase−3 significantly reduced in both the 200 and 400 mg/kg BUR groups compared with the DSS group (Figure 3H). These results indicated that BUR potentially attenuated DSS−induced apoptosis via a Bax/Bcl2/caspase 3 pathway. 

### 3.4. BUR Inhibited NLRP3 Inflammasome Activation

Considering that IL−1β and IL−18 were important NLRP3 inflammasome−dependent cytokines, and that the BUR treatment could reduce the release of these two inflammatory mediators, we then explored whether BUR inhibited NLRP3 inflammasome activation in colitis. The results showed that BUR treatment significantly decreased the mRNA and protein expression of NLRP3, ASC, and caspase−1 in DSS−treated mice (Figure 4A,B–E). Two doses of BUR produced a significant reduction in cleaved caspase 1 protein levels and blocked the maturation of IL−1β. Moreover, the mRNA and protein expression of GSDMD were obviously decreased with BUR (400 mg/kg) treatment. These results suggested that BUR could suppress NLRP3 inflammasome activation and induce pyroptosis through the NLRP3/Caspase−1/GSDMD signaling. 

### 3.5. BUR Regulated the Gut Microbial Community and Metabolites Production in Colon

The bacterial communities were tested by Venn diagram analysis, and unique organisms are listed in Figure 5A. There were 217 unique organisms found in the control group, 82 in the DSS group, 156 in the BUR (200 mg/kg) group, and 23 in the BUR (400 mg/kg) group. The number of observed species and evaluation of the alpha diversity are shown in Figure 5C–F. However, there were no differences among the four groups in terms of α−diversity. Beta diversity analysis was presented by PCoA (Figure 5G). The PCoA of OTUs revealed that the colonic microbiota of the control group were distinctly separated from the DSS and BUR groups. The top 10 most abundant bacteria at the genus level in colon are presented in Table 1. Compared with the control group, BUR supplementation markedly increased the relative abundance of *Dubosiella* and *Bifidobacterium* (*p* < 0.05). We further performed the LefSe analysis to detect distinctive bacteria between the DSS and BUR groups (Appendix A and Figure 6). The results (Appendix A) showed that Rikenellaceae, Eubacterium_siraeum_group, Rikenellaceae_RC9_gut_group, Rikenella, and Muribaculaceae were differentially enriched in the BUR (200 mg/kg) group, whereas Allorhizobium_Neorhizobium_Pararhizobium_Rhizobium, Lachnospiraceae_FCS020_group, Hydrotalea, Sphingomonadaceae, Sphingomonadales, Sphingomonas, Asinibacterium, Streptococcus, and Streptocaccaceae were differentially enriched in the DSS groups. Compared with the DSS group, the BUR (400 mg/kg) group had higher relative abundances of *Dubosiella*, *Actinobacteriota*, *Actinobacteria*, *Bifidobacterium*, *Bififobacteriales*, *Bififobacteriaceae*, *Atopobiaceae*, *Olsenella*, *Paraburkholderia*, *Burkholderiaceae*, *Tissierellales*, *Eubacterium_nodatum*, *Parvibacter*, *Coriobacteriaceae_UCG_002*, *Peptostreptococcaceae*, and *Romboutsia*. 

The concentrations of total acids including acetate, propionate, and butyrate showed a notable reduction in DSS−treated mice (Figure 7). BUR treatment significantly improved the colonic SCFAs’ profiles. BUR (400 mg/kg) treatment significantly increased the concentrations of intestinal SCFAs but only the concentrations of propionate and butyrate were remarkably improved at 200 mg/kg BUR.

### 3.6. Correlation between Gut Microbiota (at the Genus Level) and Intestinal Inflammatory Profiles

To further determine the potential relationship between gut microbiota and the inflammatory response, we performed Spearman’s rank correlation analysis between the top 10 most abundant bacteria at the genus level and the inflammatory mediators (Figure 8). The results showed that *Muribaculaceae_norank* showed significant positive correlations with serum IL−18, MPO activity, and NLRP3 protein expression. *Lactobacillus* had significant positive correlations with the acetate and total acid concentrations while it had negative correlations with serum IL−6 and IL−18, MPO activity, and ASC protein expression. The abundance of *Burkholderia−Caballeronia−Paraburkholderia* had significant positive correlations with serum IL−6 and IL−1β. *Desulfovibrio* was positively correlated with serum TNF−α and ASC protein expression.

### 3.7. Functional Prediction of Gut Microbial Community

The bio−function of gut microbial communities corresponds to their compositional changes. Thus, we performed the functional analysis of intestinal microbial community by PICRUST. As shown in Figure 9A, 26 KEGG pathways of DSS−treated mice were found to be changed as compared to those of mice in the control group. After BUR treatment, three KEGG were improved compared with the DSS group including flavone and flavonol biosynthesis, and fatty acid biosynthesis in the BUR groups (Figure 9B,C). 

## 4. Discussion

With the increasing incidence and prevalence globally, IBD has been perceived as one of the major problems threatening human health. Multiple studies have highlighted the pathogenic role of immune−mediated inflammatory disorders in IBD through mechanisms involving intestinal barrier defects, inflammasome activation, and gut microbiota dysbiosis [30]. In this study, we found that BUR supplementation alleviated DSS−induced colitis in mice by improving intestinal barrier function, inhibiting apoptosis, blocking the inflammatory signal, and modulating the gut microbial community. Importantly, the protective effects of BUR were involved in suppressing the maturation of pro−inflammatory cytokines and inhibiting the pyroptosis via NLRP3 inflammasome inactivation. 

The classical DSS−induced colitis in rodent models with morphological changes and inflammatory disorder is consistent with the clinical symptoms of IBD, including body weight loss, rectal bleeding, shortened colon length, and elevated DAI scores [31]. In the current study, we found the above−mentioned manifestations were effectively alleviated by the 200 and/or 400 mg/kg BUR supplementation. BUR pretreatment prevented the DSS−induced body weight loss, colon length reduction, and DAI score increase. In addition, the excessive release of pro−inflammatory cytokines into the systemic circulation characterizes a systemic inflammatory reaction as colitis occurs, which provides reliable indicators for the disease severity. BUR supplementation significantly reduced serum IL−6, TNF−α, IL−1β, and IL−18 to approximately the control levels. These results confirmed the protective effects of BUR in DSS−induced colitis. 

In rodents with colitis, the abnormal inflammatory cells’ infiltration is enhanced because of the intestinal barrier defects, and the goblet cells are also lost due to the disruption of the intestinal epithelial layer [32]. Our histological results showed that BUR treatment not only alleviated histopathologic damage induced by DSS, but also mitigated the goblet cells’ loss. In parallel, the reduced MPO activities in the colon were observed following BUR pretreatment, indicating the beneficial effects of BUR in the remission of colitis. The intestinal barrier represents an important physical mechanism that provides a defense against pathogens and endotoxins, which, in turn, can be linked with intestinal epithelial integrity and immune−mediated inflammatory responses [33]. In the present study, the decreases in tight junction (TJ) molecules, including occludin, claudin−1, and ZO−1, led to intestinal barrier dysfunction and promoted a strong intestinal inflammation. Curcumin and BUR were demonstrated to prevent intestinal tight junction dysfunction caused by leptin exposure, hypobaric hypoxia injury, and cisplatin poisoning [34,35,36,37]. Our results were consistent with previous studies demonstrating that BUR treatment inhibited the decreased mRNA expression of ZO−1, occludin, and claudin−1 genes caused by DSS. In BUR groups, the protein expression of ZO−1, occludin, and claudin−1 were restored as well. These results suggested that BUR treatment reduced the intestinal barrier impairment and relieved the colonic damage via regulating the TJ molecules. 

One of the remarkable findings of this study was that we determined the induction of apoptosis in colitis and BUR had an obvious protection against apoptosis. Apoptosis is a form of programmed cell death that implicated in the pathogenesis of colitis [38]. Massive intestinal epithelial apoptosis leads to the dysregulation of the intestinal barrier integrity, and thus triggers a severe inflammatory response in colitis. Our TUNEL staining demonstrated the extensive apoptosis in the DSS−treated colon. BUR supplementation decreased the mRNA expression of pro−apoptotic Bax and increased that of anti−apoptotic Bcl2. A few studies have revealed the effects of BUR on the apoptosis. Research has clarified that BUR abolished cisplatin−induced apoptosis in renal tubular epithelial cells and suppressed apoptosis induced by staurosporine and tert−butyl hydroperoxide [18,23,39]. The Bcl2 family proteins participate in the classical mitochondrial−dependent apoptotic pathway that mediates the cell death [40]. The induction of Bax leads to the release of cytochrome *c* and subsequently activates the downstream caspase 3, which degrades protein substrates to form the terminal cleavage and contributes to the DNA fragmentation [41]. By Western blotting analysis, we confirmed the inhibitory effects of BUR on the activation of caspase 3. Because of the reduction in apoptosis−executing molecules such as caspase 3 and cleaved caspase 3, BUR served as an effective apoptotic inhibitor against colitis, which might explain the mechanism by which BUR maintained the intestinal barrier integrity and blunted the DSS−induced inflammation. 

Previous studies showed that the induction of apoptosis was required for the activation of the NLRP3 inflammasome in the signaling cascade, but the concrete contribution of NLRP3 inflammasome activation to DSS−induced colitis was still elusive. There were some reports about the potential involvement of the NLRP3 inflammasome in intestinal epithelial cells after DSS−induced colitis [10,42]. In this study, BUR was found to inhibit NLRP3 inflammasome activation in DSS−treated mice. Consistent with this notion, Gong (2018) et al. found that curcumin strongly suppresses DSS−induced NLRP3 inflammasome activation and alleviates colitis in mice [14]. The less−cleaved caspase−1 and ASC specks by pretreatment produced the inhibitory effects of curcumin on the NLRP3 inflammasome activation and IL−1β secretion in LPS−primed peritoneal macrophages and BMDMs [14]. The activation of the NLRP3 inflammasome depends on two signals: priming and activation. The priming signal is achieved by the NLPR3 and IL−1β over−expression mediated by external stimuli [7]. First, our data revealed a decreased expression of NLRP3 and IL−1β in BUR groups compared with the DSS group, suggesting that BUR diminished the first signal required by the NLRP3 inflammasome activation. The activating signal requires the assembly of NLRP3 inflammasome components comprising NLRP3, ASC, and pro−caspase1, which promote caspase−1−mediated IL−1β and IL−18 secretion and pyroptosis [43]. Thus, the involvement of caspase−1 and GSDMD, a biomarker of pyroptosis, was also determined to investigate whether BUR inhibited the second assembly signal of the NLRP3 inflammasome. We found that BUR decreased the protein expression of caspase 1 and blocked the activation of caspase 1. The formation of the NLRP3 inflammasome complex requires activated caspase−1 involvement. The activation of caspase−1 cleaves GSDMD to trigger pyroptosis mediated by the mature NLRP3 inflammasome [44]. Pyroptosis leads to cell death coupled to significant increases in IL−1β and IL−18, which in turn promote the inflammatory process [45]. The results demonstrated that BUR pretreatment decreased the GSDMD protein levels, which provided another explanation for the anti−inflammatory activity of BUR related to its inhibitory role on the NLRP3 inflammasome pathway. In addition, BUR−induced apoptosis inhibition played an indirect role in the induction of NLRP3 inflammasome activation due to the diminished release of IL−1β mediated by the reduction in Bcl2 [46]. The DNA fragment, especially the damaged mitochondrial DNA, is an activator of the NLRP3 inflammasome activation [46]. We assumed that suppression of Bcl2 expression and the release of DNA fragments mediated, at least in part, the inhibitory effects of BUR on the NLRP3 inflammasome activation. The mechanisms underlying the beneficial effects of BUR were involved in both the inhibition of apoptosis and NLRP3 inflammasome activation, thus resulting in improved intestinal barrier function and reduced inflammatory injury.

The gut microbiota plays an important role in maintaining intestinal barrier function and is a sensitive contributor to the epithelium cell apoptosis and inflammation. The gut microbiota dysbiosis is responsible for the pathogenesis of IBD, and mediates the development of inflammatory reaction. The DSS challenge seriously disrupts the balance of the colon microbiome. BUR did not affect the alpha or beta diversity of the bacterial communities reduced by DSS. However, at the genus level, the abundances of multiple bacteria were rebalanced in the BUR−treated groups. *Lactobacillus* is a beneficial bacteria prevalent in the healthy intestine and depleted in IBD patients. Enriched *Lactobacillus* has been reported to improve intestinal barrier integrity and restrain pathogen growth, resulting in the increase in SCFAs producers and an anti−inflammatory response. In agreement, our HPLC analysis of colonic digesta showed that the DSS challenge significantly decreased the production of gut microbiota−produced SCFAs. The concentrations of acetate, propionate, and butyrate in the BUR groups were higher than those of the control group. There was a mutual regulation between SCFAs and gut inflammation. The enhanced SCFAs production induced by BUR supplementation might prevent the inflammation caused by DSS. Moreover, the LEfSe analysis showed that the BUR groups had higher populations of *Bifidobacterium* compared with the DSS group. *Bifidobacterium* assists in promoting gut health, and commonly serves as a potential probiotic and an alternative intervention of animal nutrition. Previous studies found that high digestive abundance of *Bifidobacterium* and *Lactobacillus* contributed to improving intestinal inflammation by suppressing NF−kappaB, and they could restrict the colonization of pathogenic bacteria by lowering gastrointestinal tract pH. The increased abundances of health−promoting taxa in the BUR groups showed the potential application of BUR in treating colitis by regulating the gut microbiota. 

Growing evidence has shown that the alteration of bacterial communities was closely associated with gut disease severity, especially the intestinal inflammatory status. Analogously, in the present study, the correlation analysis between the top 10 abundant genus bacteria and a wide spectrum of inflammation−related biomarkers was determined. The results showed that the abundance of *Lactobacillus* had a highly positive correlation with the SCFAs profiles while it had negative correlations with inflammatory factors, which were conducive to the suppression of intestinal inflammation. *Lachnospiraceae_NK4A136_group* is a kind of SCFAs−producing bacteria that promotes the biosynthesis of butyric acid [47,48]. We found that the abundance of the *Lachnospiraceae_NK4A136_group* was significantly increased in the DSS group, whereas its increase was also observed in the BUR groups. It was assumed that the improved SCFAs profiles were related to the enrichment of the *Lachnospiraceae_NK4A136_group*. *Dubosiella* may be a potentially beneficial bacteria and its enrichment helps protect in some dietary treatments against colitis [49]. Although DSS induced the increase in *Dubosiella*, we also demonstrated a similar increased abundance of *Dubosiella* in BUR groups, consistent with previous reports [50,51]. The enrichment of *Burkholderia−Caballeronia−Paraburkholderia* and *Desulfovibrio* were positively associated with the biomarkers that related to the NLRP3 inflammasome pathway. The role of *Burkholderia−Caballeronia−Paraburkholderia* in colitis is controversial. Wang (2022) et al. found that *Burkholderia−Caballeronia−Paraburkholderia* belongs to the proteobacteria phylum and is the dominant bacteria in the model mice of DSS−induced colitis, which was similar to our results [52]. Chibuzor−Onyema (2021) et al. reported that *Burkholderia−Caballeronia−Paraburkholderia* exhibits similar lactic acid−producing effects as *Lactobacillus* and *Lactococcus* [53]. An increased abundance of *Burkholderia−Caballeronia−Paraburkholderia* were also observed in the BUR groups, indicating that these conflicting results warranted further exploration. *Desulfovibrio* is a sulfate−reducing bacteria and frequently causes human infection by generating LPS [54,55]. As reported, the abundance of *Desulfovibrio* was decreased in resveratrol−treated high−fat diet mice, which was consistent with our results showing an improved gut barrier function in BUR groups [56]. The severity of the intestinal inflammatory response was decreased after BUR treatment in colitis, and the abundance of harmful bacteria (*Muribaculaceae_norank* and *Desulfovibrio*) were also decreased. BUR had a good anti−inflammatory activity and contributed to maintaining the abundances of *Bifidobacterium* and *Limosilactobacillus*. We speculated that the changed abundance of gut microbiota following BUR supplementation might avail in the regulation of inflammatory status and contribute to the remission of colitis. 

Of note, the predicted function of the changed gut microbiota were obtained with a phylogenetic reconstruction of unobserved states (PICRUSt) analysis. The function of the “two−component system” and “ABC transporters” were enriched in the DSS group compared with the control group. The ABC transporters are involved in the uptake of micronutrients and the export of cytotoxic compounds for gut microbiota [57]. BUR (200 mg/kg) supplementation enhanced the function of “fatty acid biosynthesis” and BUR (400 mg/kg) enhanced the function of “flavone and flavonol biosynthesis” in DSS−challenged mice. Based on these conflicting results of two doses, we speculated that the effects of BUR on gut microbiota were not in a concentration−dependent manner. To be honest, the results of gut microbiota are rarely consistent since distinct pathways are implicated in the regulation of natural extracts including BUR. BUR is a natural polyphenol derived from the herbal root of *Curcuma longa*. Once administrated, BUR metabolism mainly occurs via reduction and conjugation that generate various degradation and metabolism products. The assumption that natural compounds such as BUR exert a broad spectrum of biological and cellular activities directly, or indirectly through their primary and secondary metabolites, may provide an explanation for the complicated mechanism of BUR in vivo. BUR concentrations are rarely detectable in plasma at 24 h with low dietary inclusion levels, suggesting its rapid systemic metabolization and elimination. We consider that the indirect effect derived from various metabolites play a primary role in the regulation of the signaling pathway, such as fatty acid biosynthesis in the BUR (200 mg/kg) group. Increased dietary inclusion level of BUR may be related to the induction of flavone and flavonol biosynthesis because elevated BUR accumulation in serum and digesta has been observed in the BUR (400 mg/kg) group.

## 5. Conclusions

In conclusion, our study demonstrated that BUR alleviated DSS−induced colitis by improving intestinal barrier function, reducing apoptosis, and decreasing inflammation through the NLRP3 inflammasome inactivation. In addition, BUR treatment improved the gut microbial community towards exerting anti−inflammatory effects and promoting host health. These results indicated that BUR might represent a promising protective agent against intestinal damage and gut microbiome dysbiosis for the treatment and therapy of IBD.

## Figures and Tables

**Figure 1 antioxidants-11-01994-f001:**
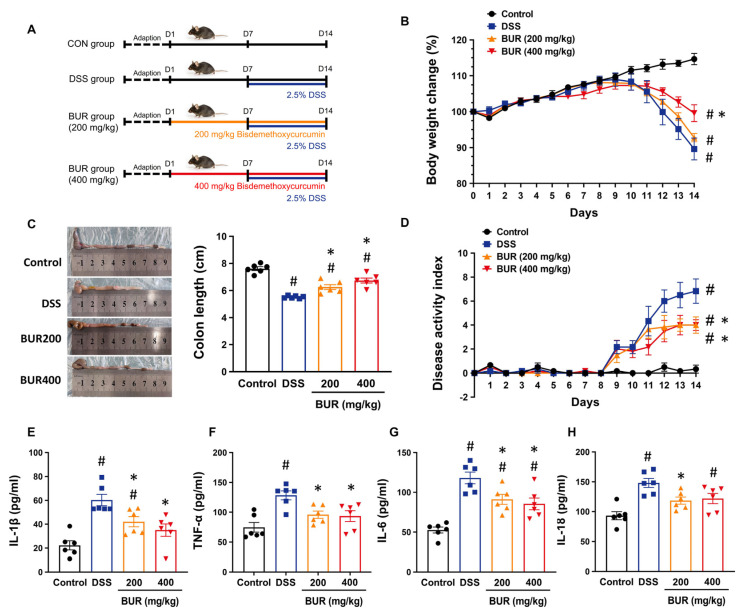
BUR treatment alleviated DSS−induced colitis in mice. (**A**) The timeline and procedure of the experiment. The colitis model was induced by administration of 2.5 % DSS in drinking water for day 7−14. BUR (200/400 mg/kg) or corn oil were administered by gavage once per day for 14 days. On day 15, mice in each group were sacrificed; (**B**) Body weight change; (**C**) Colon length; (**D**) The DAI scores; (**E**−**H**) The levels of IL−1β, TNF−α, IL−6, and IL−18 in serum. Data are presented as the mean value ± SEM (*n* = 6). # *p* < 0.05, versus the control group; * *p* < 0.05, versus the DSS group.

**Figure 2 antioxidants-11-01994-f002:**
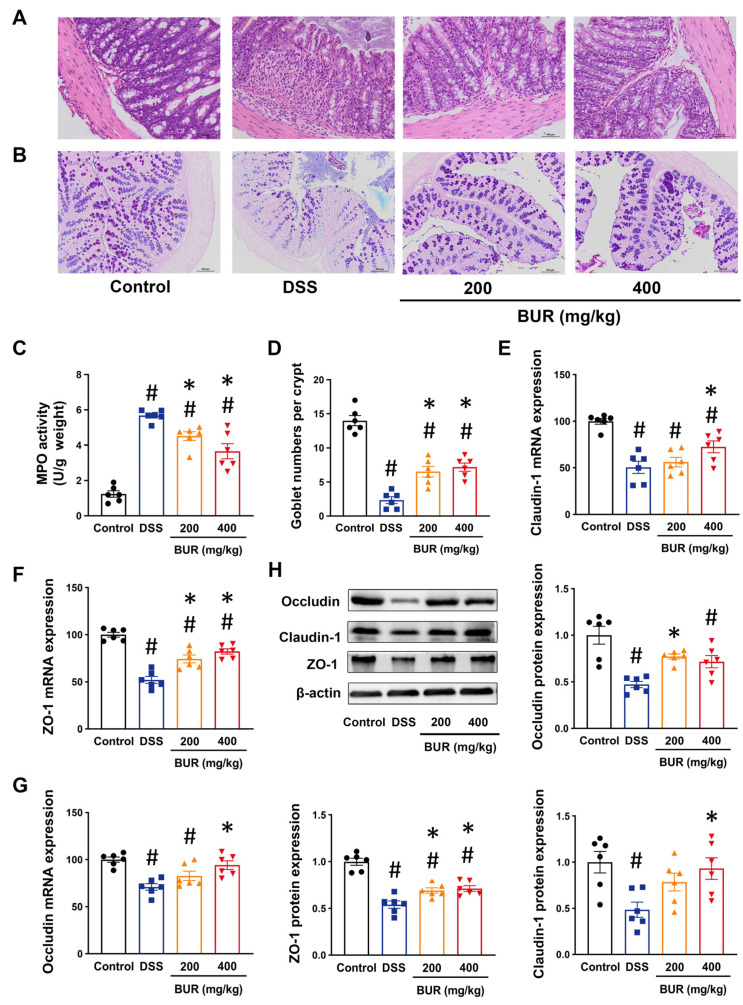
BUR relieved intestinal barrier defects in DSS−induced colitis. (**A**) Histological images of HE staining. Scale bars, 100 μm. Original magnification ×200; (**B**) Histological images of AB−PAS staining. Scale bars, 100 μm. Original magnification ×100; (**C**) Colonic MPO activity; (**D**) Goblet numbers; (**E**−**G**) The mRNA expressions of occluding, claudin−1 and ZO−1 in colon analyzed by RT−PCR; (**H**) The protein expressions of occluding, claudin−1, and ZO−1 in colon analyzed by Western blotting. Data are presented as the mean value ± SEM (*n* = 6). # *p* < 0.05, versus the control group; * *p* < 0.05, versus the DSS group.

**Figure 3 antioxidants-11-01994-f003:**
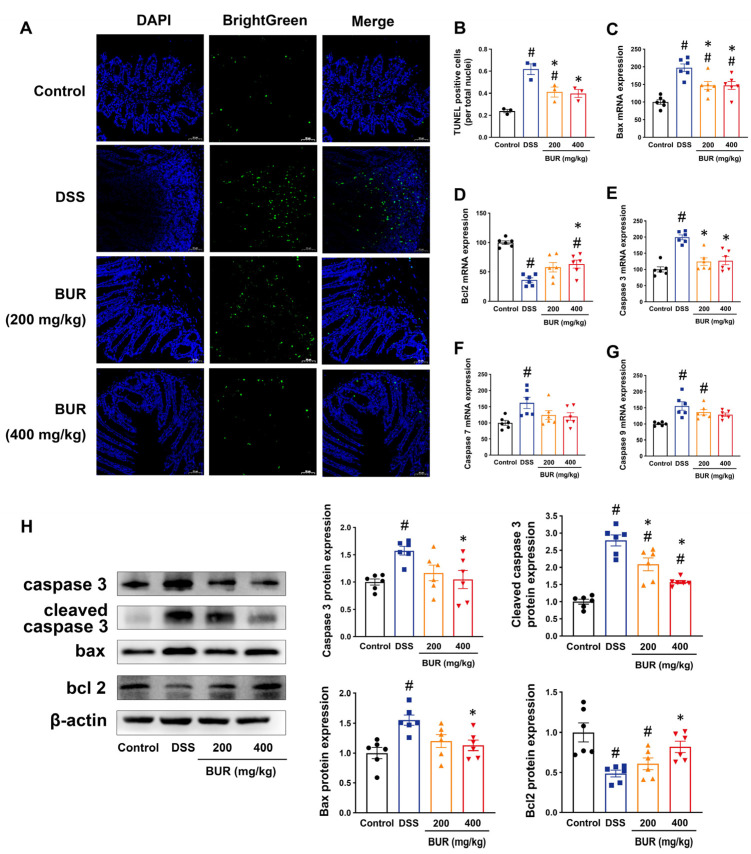
BUR suppressed intestinal epithelial apoptosis in DSS−induced colitis. (**A**) Immunofluorescence images of TUNEL staining. Scale bars, 50 μm; (**B**) The number of TUNEL−positive cells in colonic sections (*n* = 3); (**C**−**G**) The mRNA expressions of Bax, Bcl2, caspase 3, caspase 7, and caspase 9 analyzed by RT−PCR; (**H**) The protein expressions of caspase 3, cleaved caspase 3, Bax, and Bcl2 analyzed by Western blotting. Data are presented as the mean value ± SEM (*n* = 6). # *p* < 0.05, versus the control group; * *p* < 0.05, versus the DSS group.

**Figure 4 antioxidants-11-01994-f004:**
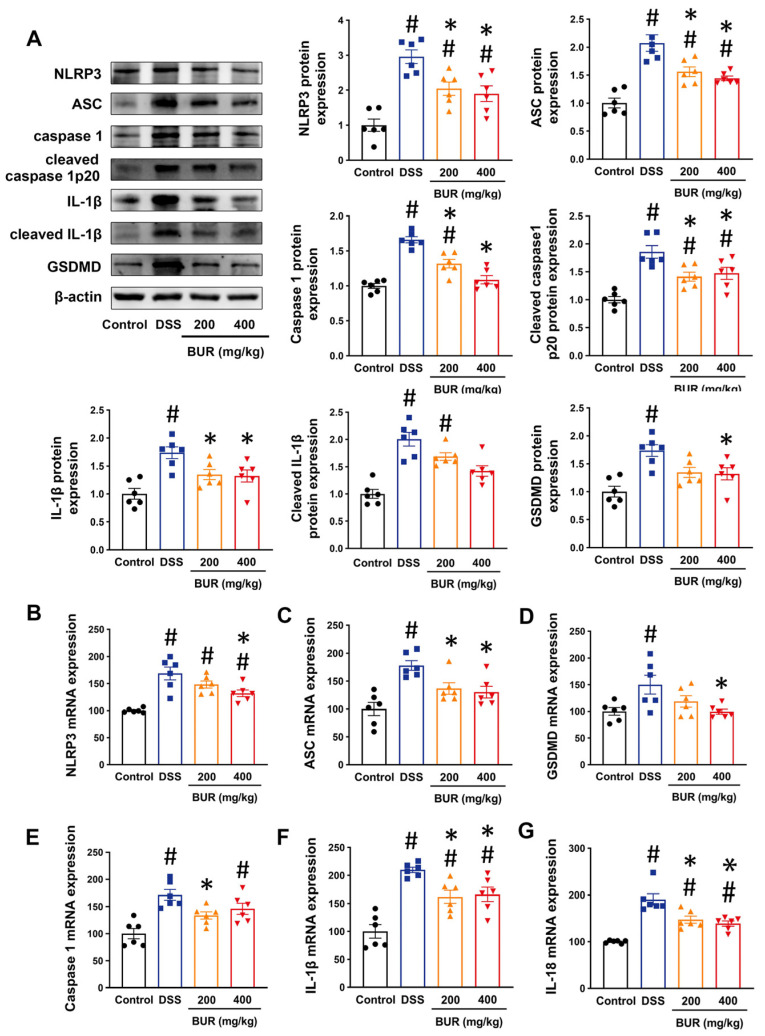
BUR inhibited NLRP3 inflammasome activation in DSS−induced colitis. (**A**) The protein expressions of NLRP3, ASC, caspase 1, cleaved caspase 1, IL−1β, cleaved IL−1β, and GSDMD analyzed by Western blotting; (**B**−**G**) The mRNA expressions of NLRP3, ASC, GSDMD, caspase 1, IL−1β and IL−18 analyzed by RT−PCR. Data are presented as the mean value ± SEM (*n* = 6). # *p* < 0.05, versus the control group; * *p* < 0.05, versus the DSS group.

**Figure 5 antioxidants-11-01994-f005:**
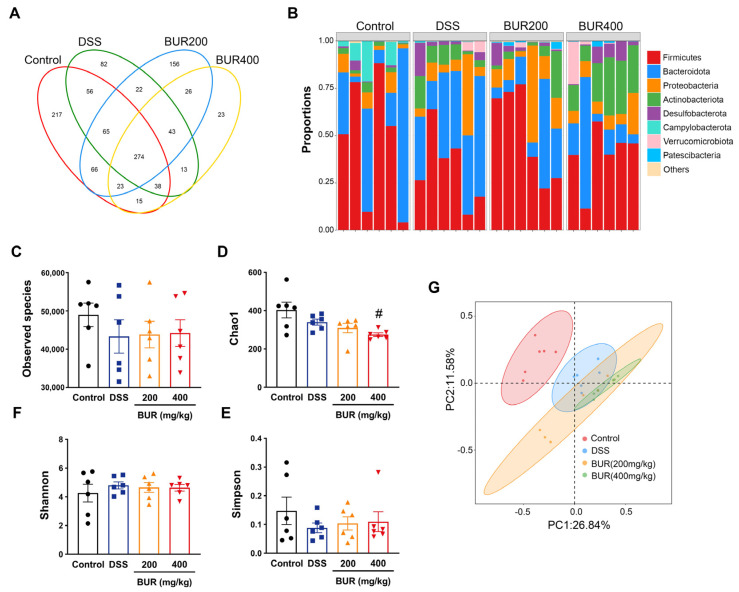
BUR altered the gut microbial structure in DSS−induced colitis. (**A**) Venn diagrams of OTUs; (**B**) The relative abundances of gut microbial community at the phylum level; (**C**−**F**) alpha diversity as measured by the Chao1, Simpson, and Shannon index at OTUs level; (**G**) beta diversity presented by PCoA. Data are presented as the mean value ± SEM (*n* = 6). # *p* < 0.05, versus the control group.

**Figure 6 antioxidants-11-01994-f006:**
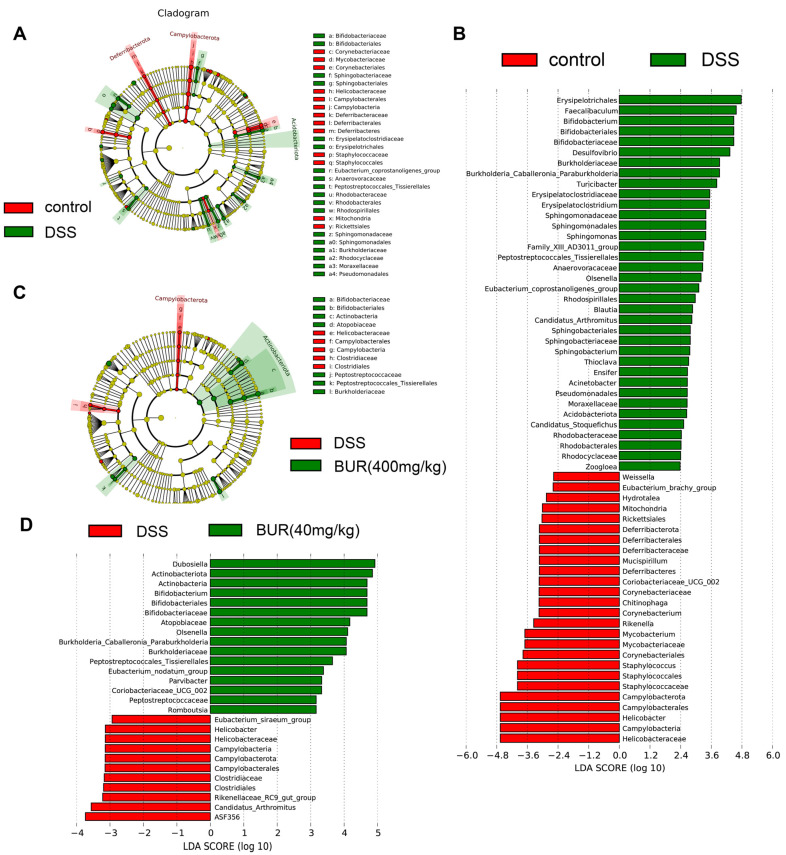
LEfSe analysis of the gut microbial composition among treatment groups (LDA > 2, *p* < 0.05). (**A**) Cladogram representation of the microbiota taxa between the control group and DSS group; (**B**) Cladogram representation of the microbiota taxa between the DSS group and BUR (400 mg/kg) group; (**C**) LDA of the microbiota taxa between the control group and DSS group; (**D**) LDA of the microbiota taxa between the DSS group and BUR (400 mg/kg) group.

**Figure 7 antioxidants-11-01994-f007:**
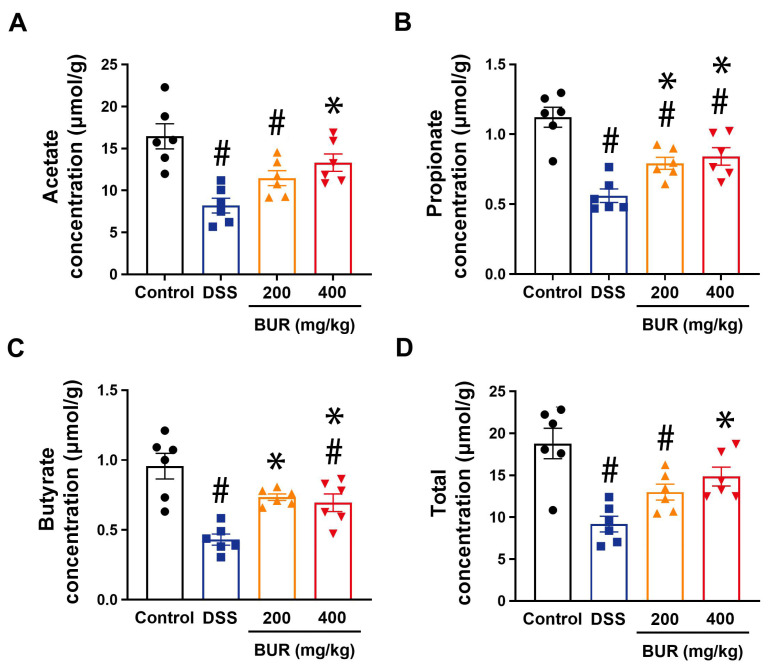
BUR improved the production of SCFAs in DSS−induced colitis. (**A**) Concentrations of acetate in colonic digesta; (**B**) Concentrations of propionate in colonic digesta; (**C**) Concentrations of butyrate in colonic digesta; (**D**) Total concentrations of the acetate, propionate and butyrate. Data are presented as the mean value ± SEM (*n* = 6). # *p* < 0.05, versus the control group; * *p* < 0.05, versus the DSS group.

**Figure 8 antioxidants-11-01994-f008:**
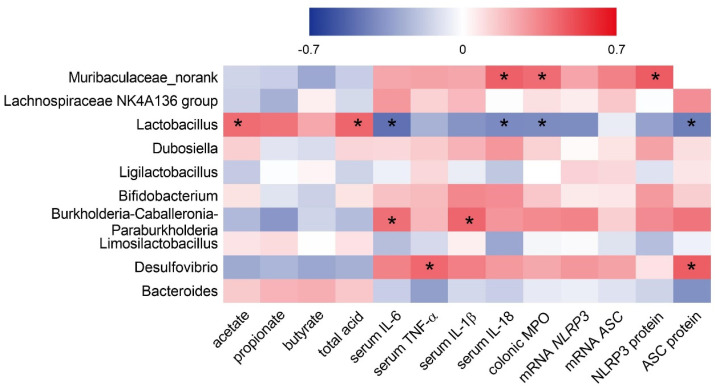
Heatmap of the correlation between the relative abundance of top 10 bacteria and inflammatory−related mediators by the Spearman’s rank correlation test. The red color indicates the positive correlation while the blue color indicates the negative correlation. * *p* < 0.05.

**Figure 9 antioxidants-11-01994-f009:**
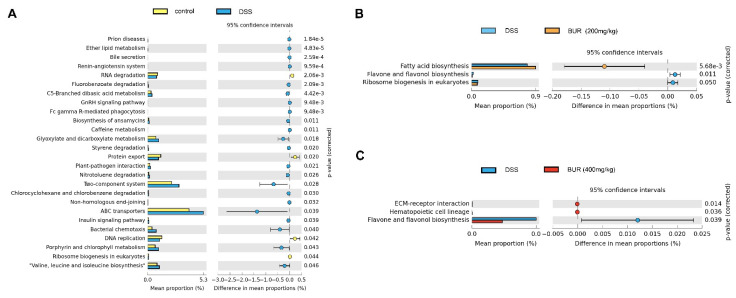
Functional prediction of gut microbial community among treatment groups. By PICRUSt analysis, the significantly enriched KEGG pathways (at level 3) were determined by STAMP software. (**A**) The control group versus the DSS group; (**B**) The DSS group versus the BUR (200 mg/kg) group; (**C**) The DSS group versus the BUR (400 mg/kg) group.

**Table 1 antioxidants-11-01994-t001:** The top 10 most abundant bacteria at the genus level.

Items	Control	DSS	BUR Treatment		SEM	*p*−Value
			200 mg/kg	400 mg/kg		
*Muribaculaceae_norank*	20.93	36.80	10.51	16.24	4.21	0.140
*Lachnospiraceae NK4A136 group*	2.22	9.47	19.89	7.06	2.96	0.190
*Lactobacillus*	25.01	0.20	0.17	0.21	3.78	0.031
*Dubosiella*	0.39	2.90	1.99	19.58 ^#^*	1.96	<0.0001
*Ligilactobacillus*	8.11	2.90	12.59	1.07	1.87	0.110
*Bifidobacterium*	0.04	6.05	3.07	15.07 ^#^*	1.52	<0.0001
*Burkholderia−Caballeronia−Paraburkholderia*	0.10	1.77	7.39	4.58	1.12	0.094
*Limosilactobacillus*	5.84	1.13	5.20	1.27	1.10	0.287
*Desulfovibrio*	0.56	4.46	2.63	4.75	0.93	0.369
*Bacteroides*	4.84	0.64	2.40	1.36	0.84	0.321

Data are presented as the mean value ± SEM (*n* = 6). # *p* < 0.05, versus the control group; * *p* < 0.05, versus the DSS group.

## Data Availability

The data used in the present study are available on request from the corresponding author.

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
