# Peer review of "Bisdemethoxycurcumin Alleviates Dextran Sodium Sulfate-Induced Colitis via Inhibiting NLRP3 Inflammasome Activation and Modulating the Gut Microbiota in Mice"

_antioxidants, 2022, doi:10.3390/antiox11101994_

Round 1

Reviewer 1 Report

Dear Editor,

The study submitted by Zhang J and collaborators entitled ‘Bisdemethoxycurcumin alleviates dextran sodium sulfate-induced colitis via inhibiting NLRP3 inflammasome activation and modulating the gut microbiota in mice’ aimed at exploring the ameliorating effects of a derivative of curcumin with improved pharmacological properties, bisdemethoxycurcumin, on experimentally induced DSS-induced colitis in mice. In this study, the authors studied in great detail the underlying mechanisms implicated in the beneficial effects of BUR. The results demonstrated that the prophylactic administration of BUR to mice could attenuate body weight loses, DAI, which was confirmed histopathological analysis of colons. Several mechanisms of action are implicated in the significantly reduced susceptibility of DSS/BUR-treated mice with respect to DSS-treated mice. Among those, reduced intestinal epithelial cell apoptosis, together with reduced expression of NLRP3 inflammasome activation and pyroptosis thus resulting in a better preserved intestinal barrier and reduced inflammatory injury in DSS/BUR-tretated mice with respect to DSS-treated mice. Additionally, the authors demonstrated that BUR contributed to the maintenance of Bifidobacteria and Limosilactobacillus. that inversely correlated with the inflammatory status of DSS colitic mice. Moreover, increase levels of SCFAs were observed in DSS/BUR-treated mice when compared to DSS-treated colitic mice which may explain the competitive advantage of probiotic bacteria upon administration of BUR during induction of DSS colitis.

The data provided by the authors are robust and compiling and I recommend the publication of the study once the authors correct some minor points as described in detail below:

1.       The BUR was administered prophylactically and during the DSS treatment so the BUR did not formally reverse the experimental colitis. The curcumin administration should be  considered as a prevented treatment rather than a curative procedure. Throughout the manuscript the authors used the term ‘reversed’, or ‘induced down-regulation’. Instead, the terms ‘prevented’, or ‘reduced induction of inflammasome…’ should be used.

2.       Increase the font sizes and size of pictures in figures 1C; 5C; 6 A to D to make the text readable.

3.       Pictures shown in figure 3A are too dark to see anything clearly. Please, improve the quality of the images chosen as representative examples.

4.       Include the scale bars in figure 2 A and 2B.

5.       The statistical analysis seem correct, however, the use of letter such as ‘a’, ‘b’, and so on is quite confusing for the reader as the treatment groups that are compare under each letter are nor clearly depicted. Please, use a more conventional style for showing the groups that are compared in each figure.   

6.       Was the overall survival of mice altered in any of the treatments, particular in the DSS treatment group?. The authors do not comment anything related to the survival of mice during DSS treatment, which may be of interest in case there were differences among treatments. May be a brief comment in results can be included.

7.       In a previous study published in 2018 (Reference 14), it was shown that curcumin administration significantly alleviated DSS-induced colitis via inhibiting NLRP3 inflammasome and IL1b secretion. The authors should discussed these previous findings in the Discussion.  

8.       In the last paragraph of the manuscript the authors claimed that ‘…enhanced SCFAs production and colonization of probiotic bacteria were beneficial for the defense of DSS challenge and the promotion of gut health in mice’. Formally the data presented in the study do not directly address this point, it rather seem an overinterpretation of their findings as data from healthy mice treated with BUR alone are not shown. Please correct that in the revised version of the manuscript.

9.   In the Introduction or in the Discussion the authors should comment more extensively the advantages of using ‘bisdemethoxycurcumin’ instead of the native form of ‘Curcumin’.

Reviewer 2 Report

The manuscript was reviewed for publication in the journal. The manuscript was designed to evaluate the effect of bisdemethoxycurcumin (BUR) in DSS-induced colitis via inhibiting NLRP3 inflammasome activation and modulating the gut microbiota dysbiosis. The results obtained show that BUR alleviate DSS-induced colitis by improving intestinal barrier function, reducing apoptosis, decreasing inflammation through the NLRP3 inflammasome inactivation. It is the reviewer’s opinion that the manuscript is interesting and easy to follow. However, it appears that there are a couple of concerns in the manuscript.

1) BUR was used at a dose of 200 or 400 mg/kg in the study. How did the authors decide the dosage? Based on your previous study (literature: No. 19), or other previous study? The authors should explain the point.

2) DSS and BUR were both administered orally. In some case, oral administration of material appears to reduce DSS induction. How about the drinking volume of DSS? The authors should explain the issue.

3) The authors mentioned that BUR alleviate DSS-induced colitis by decreasing inflammation through the NLRP3 inflammasome inactivation. However, the authors showed that BUR inhibited NLRP3 inflammasome activation in Figure 4. The authors should use NLRP3 activator to cancel the protective effect of BUR in DSS-induced colitis.

4) Figure 2A, 2B, and 3A include scale bars. They appear to be difficult to see. 

5) In terms of intestinal barrier function, expressions of tight junction proteins such as ZO-1, claudin-1, and occludin were only evaluated. How about other evaluation of intestinal barrier function such as intestinal permeability and cell experiment? The authors should discuss the issue.

6) 400 mg/kg of BUR appears to attenuate DSS-induced colitis compared to 200 mg/kg of BUR based on the results in Figure 1, 2, 3, and 4. However, there are a few differences of gut microbial structure and the top 10 most abundant bacteria as shown in Figure 5 and Table 1. How did the authors speculate? The authors should discuss the issue.

7) How about LEfSe analysis of the gut microbial composition between the DSS group and BUR (200 mg/kg) group? 

8) BUR (400 mg/kg) group showed the significant increases in some genes (Dubosiella and Bifidobacterium) as shown in Table 1. However, these increases were not found in BUR (200 mg/kg) group. The authors should explain the issue.

9) Figure 7 showed that BUR improved the production of SCFAs in DSS-induced colitis. How about the results of specific bacteria that produces SCFAs? The authors should explain/discuss the issue.

10) As shown in Figure 9B and 9C, functional predictions of gut microbial community between the DSS group versus the BUR (400 mg/kg) was quite different compared to those between the DSS groups versus the BUR (200 mg/kg). How did the authors explain the point?

Reviewer 3 Report

The authors continue the work undertaken earlier on the curcumin derivative Bisdemetoxycurcumin (BUR) due to its better bioavailability in vivo. They set out to determine how BUR reduces IL-1β production by inhibiting the activation of the NLRP3 inflammasome.

Presented article is very interesting and sounds very good, it is continuation of previous studies.

I have only minor suggestions about using some words like "minification".

Please check also if it should be  "inflammsome" or " inflammasome" (with "a" in the middle)

Round 2

Reviewer 2 Report

The manuscript was re-reviewed for publication in the journal. The manuscript was designed to evaluate the effect of bisdemethoxycurcumin (BUR) in DSS-induced colitis via inhibiting NLRP3 inflammasome activation and modulating the gut microbiota dysbiosis. The results obtained show that BUR alleviate DSS-induced colitis by improving intestinal barrier function, reducing apoptosis, decreasing inflammation through the NLRP3 inflammasome inactivation. It is the reviewer’s opinion that the manuscript is quite interesting and easy to follow. The authors promptly explained/discussed all issues/points. I have no more concern in the manuscript.